# Effects of Visual Distinctiveness on Learning and Retrieval in Icon Toolbars

Febi Chajadi*        Md. Sami Uddin†        Carl Gutwin‡

University of Saskatchewan

## ABSTRACT

Learnability is important in graphical interfaces because it supports the user's transition to expertise. One aspect of GUI learnability is the degree to which the icons in toolbars and ribbons are identifiable and memorable – but current "flat" and "subtle" designs that promote strong visual consistency could hinder learning by reducing visual distinctiveness within a set of icons. There is little known, however, about the effects of visual distinctiveness of icons on selection performance and memorability. To address this gap, we carried out two studies using several icon sets with different degrees of visual distinctiveness, and compared how quickly people could learn and retrieve the icons. Our first study found no evidence that increasing colour or shape distinctiveness improved learning, but found that icons with concrete imagery were easier to learn. Our second study found similar results: there was no effect of increasing either colour or shape distinctiveness, but there was again a clear improvement for icons with recognizable imagery. Our results show that visual characteristics appear to affect UI learnability much less than the meaning of the icons' representations.

**Index Terms:** Human-centered computing—Human computer interaction (HCI); Human-centered computing—User Interface Design

## 1 INTRODUCTION

Learnability is important in graphical user interfaces because it is an important part of a user's transition from novice to expert. Many kinds of learning can occur with an interface, but for WIMP interfaces (systems with windows, icons, menus, and pointers), one main way that users improve their performance is by learning the commands associated with icons in toolbars and ribbons, and where those icons are located. Therefore, a goal in the visual design of icons is to help the user remember the icon and the underlying command. However, other goals in icon design may interfere with an icon's ability to communicate its intended meaning to the user. One of these goals is the desire for visual consistency and cohesiveness – the idea that all of the icons in an interface should repeat the same visual variables (such as colour, contrast, weight, shape, angle, and size) in order to tie together the visual elements of the interface and give the system a recognizable style. For example, Figure 1 shows icons presented as good examples of visual consistency in icon design. These icons also illustrate a second design goal that is common in many commercial systems – subtle and "flat" icon design, in which icons are monochrome and have relatively low contrast.

Although these types of icons are popular, the similarity across several visual variables also reduces distinctiveness, which could hinder visibility, learnability, and memorability (in the limit, if all of a UI's icons were identical grey rectangles, they would be difficult

---

*e-mail: febi.chajadi@usask.ca
†e-mail: sami.uddin@usask.ca
‡e-mail: gutwin@cs.usask.ca

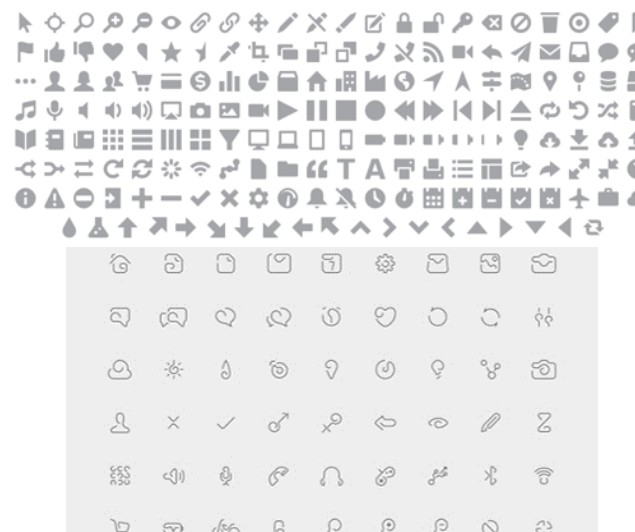

Figure 1: Two examples from web posts on "visually cohesive icon design" [27] and "minimalist icon design" [70]. Visual variables such as colour, contrast, weight, and size are repeated across the entire set.

to remember). More generally, it seems likely that icon learnability could be affected by the visual attributes of the icons. This issue has been raised by some users who have noticed the potential problems of "flat" icon design: for example, forum posts often complain that icons are too similar (Figure 2). Recently, Microsoft has moved away from flat icons to more colourful and non-uniform imagery reminiscent of earlier guidance on graphical design [43].

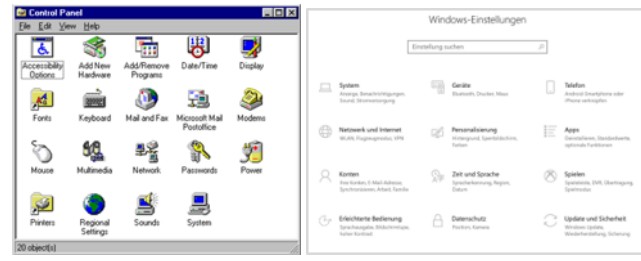

Figure 2: Blog post: "1995, SVGA: colourful and distinguishable icons. 2019, 4K and millions of colours: flat monochrome icons"

Previous research into visual variables suggests an increase in both noticeability and memorability when multimedia objects employ variations in colour and shape [8, 67, 74]. However, there are limits to how shape may contribute to better learnability. For example, an icon that cannot clearly represent its command (e.g., icons for abstract or complex commands such as "Analyse" or "Encoding") may cause confusion and impede learning despite the benefits of using representative icons [25]. Designers need to know whether the visual properties of icons can affect learnability and memora-

bility. Although there is information about the usability of certain visual properties (e.g., suggested contrast ratios between text and background [4]), there is little known about effects on learning.

To address this gap, we carried out two studies to compare how quickly people could learn and select icons with varying degrees of visual distinctiveness. Our first study tested the effects of meaningfulness in the icon's representation (comparing abstract and concrete imagery) and the effects of colour (comparing monochrome icons to icons of different colours). We also tested a fifth icon set that had substantial variation in both shape and colour. Participants were asked to find and select target commands from a toolbar with 60 icons, repeated over five blocks. The results of our first study were:

- Icons with concrete imagery were learned much faster than abstract icons;
- Adding colour to either the concrete or the abstract icon set did not lead to improved learning;
- Varying both shape and colour did not improve learnability;

Our second study tested the effects of three factors in a series of planned comparisons. We assessed meaningfulness (icons were either identical squares or concrete images), familiarity (concrete icons were either unfamiliar shapes or familiar images), and colour (squares could be either monochrome or coloured). Results from our second study were:

- The addition of colour to the identical grey squares did not improve learning;
- Unfamiliar shapes (Chinese characters for users with no familiarity with them) were much harder to learn than familiar shapes (everyday objects);
- There was no difference in learning between the Chinese characters and the grey squares, even though the characters were far more differentiable in terms of shape.

Our studies provide new information about how visual distinctiveness affects icon learning and retrieval. Surprisingly, the low visual distinctiveness of "flat" and subtle icon designs does not appear to make them more difficult to find or remember. Instead, having a concrete visual representation in the icon was shown to be extremely valuable for learning the icons. Based on our participants' comments, we suggest that this property better allows users to create a "memory hook" for the association between the icon and the command. Our studies contribute to a better understanding of how visual variables affect the process of learning icon locations, and provides a clear suggestion to designers that concrete images are likely to be more important than other forms of visual distinctiveness.

## 2 RELATED WORK

### 2.1 Visual Distinctiveness in Graphical Icons

Since the invention of GUIs, researchers have studied icon design and have identified features that can be divided into two broad categories: visual and cognitive [53]. Visual features of icons include colour, size and shape. *Colour* is one of the most prominent visual traits that can easily separate an icon from another. Despite our ability to see a huge number of colours, however, most people can only differentiate and remember about five to eight colours in a visual workspace [68]. One of the main uses for colour in interactive systems is in highlighting items, e.g., when searching [12, 14]. *Size* is another visible feature that makes icons distinguishable. Although a common use of the size feature is to make an interface cohesive (e.g., similar-sized icons used throughout a GUI; Figure 1), changing the size of an icon can make it distinct (e.g., MS Office [48] uses multiple sizes of icons). Besides colour and size, the *shape* of an icon is a strong visual factor that represents the underlying meaning [53]. Shapes can make icons more easily discernible as people can identify far more shapes than colours [69].

Cognitive features of icons are related to people's cognition and memory. Researchers mainly focus on five subjective aspects of icons: familiarity, concreteness, complexity, meaningfulness, and semantic distance [47] (see Ng et al.'s [53] and Moyes et al.'s [52] reviews for comparative summaries). Among these features, however, the success of making an icon distinguishable depends on how naturally it can depict its underlying function [10]. Although these visual variables have been studied extensively, less is known about how they contribute to learning and retrieval of icons in GUIs.

### 2.2 Psychology of Learning and Retrieval

Learning and recall are two natural yet powerful human abilities. Researchers in psychology have extensively studied human memory [5, 6, 13, 20, 21] and explored how these learning and retrieval skills are developed [3, 56, 72]. Siegel et al. [65] suggested that a combination of landmark, route, and survey knowledge contributes to the development of spatial learning and retrieval skills. People naturally begin learning objects in a new area through visual inspection [36] – which forms *landmark knowledge* [24]. Once familiar with the area, people build *route knowledge* [73] and start retrieving objects from already learned locations. With further experience, people acquire *survey knowledge* – where they can recall objects solely from memory, without requiring any visual search.

Anderson [3] and Fitts et al. [26] suggested that learning and retrieving occur in three stages: cognitive, associative and autonomous. These stages of skill acquisition can be observed in GUIs. First, in the *cognitive stage*, users learn the contents of an interface and visually look for commands. Second, in the *associative stage*, users already know the contents of the interface and begin to remember the commands in the UI. As a result, they can reach those locations more quickly. However, in the associative stage, users still perform local visual search after reaching the vicinity of a command. Last, in the *autonomous stage*, users can recall a command's location from memory and visit it, without searching for it visually.

### 2.3 Facilitating Learning and Retrieval of Icons in GUIs

Although learnability (the idea of making an interface easily learnable and memorable) is frequently considered as a vital part of usability [22, 54, 64], learnability is difficult to predict and measure [31, 61]. In order to facilitate the learning of icons and commands in GUIs, researchers have followed two main strategies: *spatial interfaces* and *landmarks*.

*Spatial interfaces.* Spatial memory [42, 56] is a human cognitive ability responsible for learning and remembering the locations of objects and places. Researchers have tried to exploit it by laying out interfaces in ways that are spatially stable [16, 23, 62]. For example, Scarr et al.'s [58, 60] CommandMap showed a spatially stable icon arrangement in desktops, yielding better learning and recall of icons, even for real tasks [59], because users could leverage spatial memory [17, 58, 78]. Similarly, Gutwin et al. [32] and Cockburn et al. [15] showed that a stable layout composed of all commands can increase recall efficiency compared to hierarchical ribbons or menus. Similar to desktops, spatially stable icons can improve learning and recall in multi-touch tablets [30, 33, 34, 77], smartwatches [45], smartphones [83, 84], digital tabletops [80], and even in VR [28].

*Landmarks in GUIs.* Landmarks are easily identifiable objects and features that are different from their surroundings [46] and which can act as anchors for performing spatial activities such as navigation and object learning. Similar to their benefits in real life, landmarks have exhibited potential in GUIs [2, 75, 76]. Researchers have exploited landmarks that are already present in the GUI environment: for example, the corners of a screen [34, 80] or the bezel of a device [63] can provide strong landmarks for icons near those locations. However, these natural landmarks often become useless in large interfaces (e.g., the middle area of a large screen) or a GUI with a large number of icons, because no landmark is present near

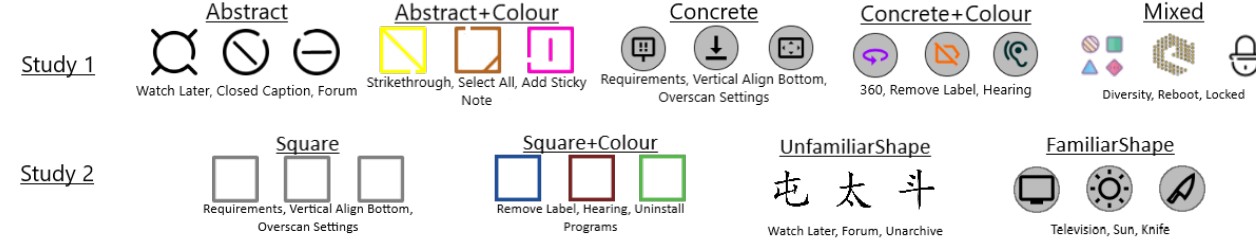

Figure 3: Example target words in each interface.

those icons. In such cases, 'artificial landmarks' [29, 78] can aid learning and recall. Studies suggested that coloured blocks [2, 78], images as the background of a menu [78] and meaningful yet abstract icons [50, 79] can be landmarks in GUIs to benefit spatial memory development.

Apart from spatial memory and landmarks, researchers have also studied features such as luminance [51] and the 'articulatory distance' [9] of icons for learnability. Others found icons representing accurate underlying meaning beneficial for learning and recall [7, 25, 39, 44, 57, 66]. Studies have shown that abstract and ambiguous icons demand more cognitive processing to recognize [40] and often hinder users from quickly learning them. However, the primary question – how visual variables of icons impact learning and recall – remains unanswered.

### 2.4 Visual Distinctiveness of Icons

We carried out two studies to investigate the effects of visual distinctiveness of icons on learnability. We manipulated two visual variables – *shape* and *colour* – but because the shape of an icon can also be representational, we also consider the cognitive variable of *meaning* in our studies as well [10, 53].

**Meaning:** The meaning of an icon refers to the concept or idea that the icon's image conveys. Icons in our studies varied by the types of underlying meaning they possess:

- *Meaningless:* Icon images have no connection to real-world objects or to their underlying commands (e.g., a grey square for the command "Settings").
- *Contextual:* Icon images are representational of underlying commands, but require interpretation if unfamiliar (e.g., a summation symbol for the command "Formula").
- *Familiar:* Icon images are pictorial and match their underlying command (e.g., an image of a calculator for the command "Calculator").

**Shape Distinctiveness:** Shape distinctiveness, that is, separability of shapes, is difficult to define precisely. Prior work has explored aspects of this concept: Julész [37] identified shape features that early visual systems detect, Burlinson et al. [11] proposed that open or closed shapes influence perceptual processing, and Smart et al. [67] investigated the perception of filled, unfilled and open shapes in scatterplots. In this work, we define levels of shape separability with respect to trends observed in modern icon design.

- *None:* Icons have no differences in shape (e.g. icons are all identical circles).
- *Medium:* Icons use different shapes, but are thematically uniform for visual consistency (similar sizes, weights, line styles, and borders).
- *High:* Icon shapes are distinctly different from one another.

**Colour Distinctiveness:** We consider only basic levels of colour distinctiveness, because people's ability to distinguish colours is much lower than the ability to distinguish shapes [69].

- *None (Monochrome):* All icons use only a single colour.
- *Medium (Colour):* Different icons use different colours.
- *High (Multi-colour):* Icons use several different colours.

Both the visual and cognitive variables operate in the context of the spatial arrangement of the icons – the two studies reported in the following sections investigate how different combinations of the levels and factors above (summarized in Table 1) affect users' ability to learn the spatial location of the icon corresponding to each command.

## 3 STUDY 1 METHODS

### 3.1 Interfaces

We developed five custom web-based desktop icon selection interfaces, each consisting of sixty icons (44 px in size) arranged in three equal rows and presented in a standard ribbon-toolbar structure. All icon toolbars appeared at the top of the interface and allowed two types of mouse-based interaction: selection and hover. Names of the icons were not shown in the UI, but could be seen in a tooltip after hovering the mouse over an icon for 300ms. Icons were created using the GIMP image editor, using source images from freely-available icon sets such as material.io and icons8. We used five experimental interfaces in Study 1, described below and shown in Figure 4.

**Concrete.** The *Concrete* interface used monochrome icons similar to those found in standard mobile and desktop environments. The icons were chosen to avoid images of real-world objects, and therefore had contextual meaning. Although icons varied in shape, the level of distinctiveness was reduced by adding a circular grey background with a 1-pixel black border.

**Concrete+Colour.** The *Concrete+Colour* interface used icons similar to *Concrete* in terms of shape distinctiveness and meaning (no icons were repeated). Icons were given a colour from a set of twelve unique colours; colours were equally distributed among the 60 icons. Colour brightness was adjusted to make icons with different colours clearly differentiable, and colours were not repeated for neighboring icons. The addition of colour provides the user with new landmarks that could be valuable for remembering locations (e.g., "it was the blue icon next to the red icon").

**Abstract.** The *Abstract* interface used meaningless monochrome icons consisting of circle and octagon shapes that were augmented with partial or full crossing lines, gaps in the outline, or dots in the centre of the icon. Icons in this set provided medium shape distinctiveness: each shape was different, but the set shared several basic visual properties.

**Abstract+Colour.** The *Abstract+Colour* interface used icons that were similar in design to *Abstract*, but used a square base outline. Colours were added to icons as described above for the *Concrete+Colour* interface.

**Mixed.** The *Mixed* interface used icons with high shape distinctiveness (variations in size, shape, weight, and texture) and high colour distinctiveness (icons used a variety of colours). These variations provide users with two different types of landmark to assist

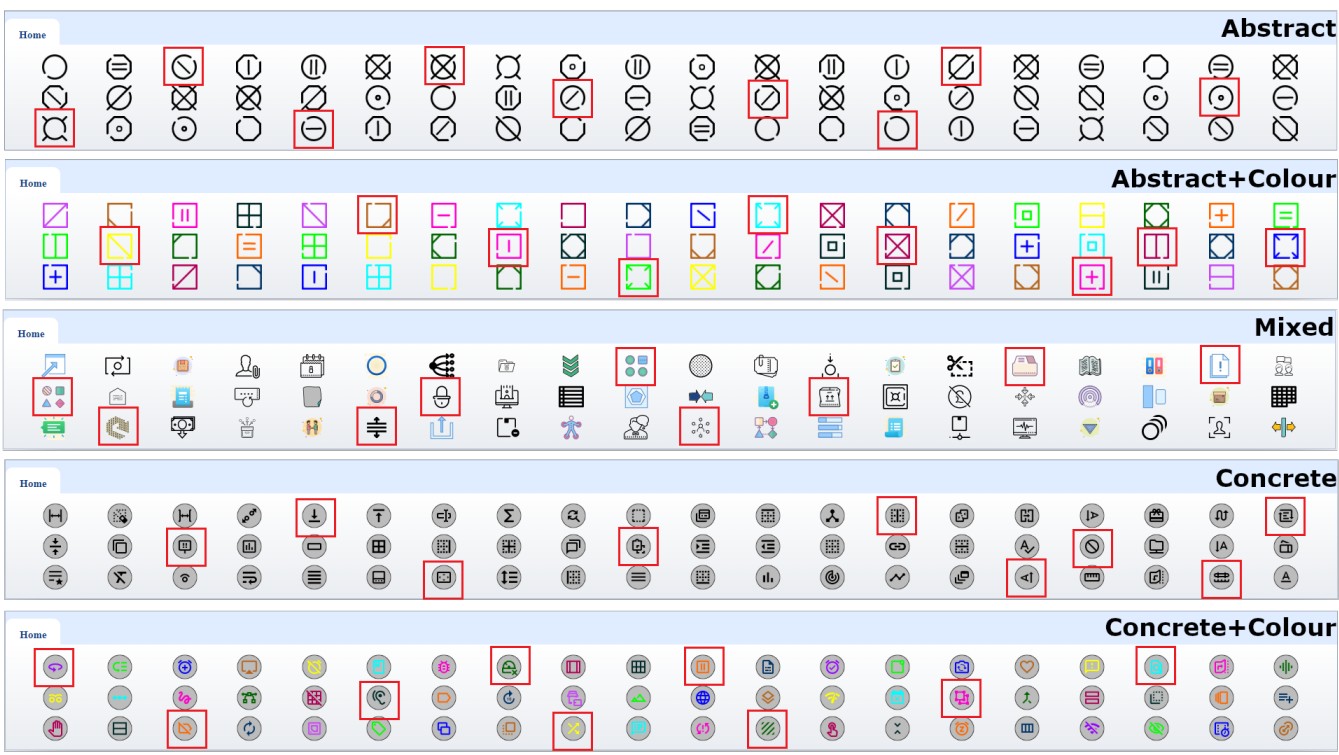

Figure 4: Screenshots of the five interfaces used in Study 1. Target objects are outlined in red.

their location memory. The icons were adapted from a real-world set, and had contextual meaning.

### 3.2 Tasks and Stimuli

The study consisted of a series of trials in each of the five interfaces, where each trial involved locating and selecting an icon. This task is commonly and frequently done in several toolbar-based or ribbon-based interfaces, such as Microsoft Word 2007 [48], Adobe Photoshop [1], or the GIMP graphics editor [71]. Every trial began by displaying a target word cue in the middle of a screen that remained visible for the entire trial, and participants were asked to find and select the corresponding icon from the toolbar. Participants could see the name of an icon as a tooltip after hovering over it for 300 ms. Each correct selection was indicated by a green flash at the selected location; red flashes were used to indicate incorrect selections. After selecting the correct icon, participants could proceed to the next trial by clicking on a 'Next Trial' button that appeared in the middle of the screen. The button centred a participant's gaze and cursor position, and started the timer of a trial. For each interface, 9 out of the 60 icons were used as targets; these were sampled from three general areas of the toolbar [78]: 3 from the corner regions (first and last three columns), 4 from the edges (top and bottom rows) and 2 from the middle row. No target position was repeated among the five interfaces. Target positions in each interface was repeated across all participants in random order of appearance.

### 3.3 Participants and Apparatus

Twenty participants (ten men, nine women, one non-binary), ages 20-44 (mean 26, SD 5.4), were recruited from a local university and received a $15 honorarium. All participants had normal or corrected-to-normal vision, and none reported a colour-vision deficiency. All participants were highly familiar with desktop and mobile applications (up to 10 hrs/wk (3), 20 hrs/wk (4), 30 hrs/wk (1) and over 30 hrs/wk (12)). The study took 90 minutes. Ten participants reported

primarily issuing commands by navigating GUIs with mice and ten reported using keyboard shortcuts. Overall participants were familiar with keyboard shortcuts (1-5 shortcuts (7), 6-10 shortcuts (9), 11-15 shortcuts (2), 16-20 shortcuts (1), and over 20 shortcuts (1)).

Study software (used in Study 1 and 2) was written in JavaScript, HTML and CSS, and ran in the Chrome browser. The study used a 27-inch monitor at 1920x1080 resolution, running on a Windows 10 PC with an Nvidia GTX 1080Ti graphics card. The system recorded all performance data; subjective responses were collected with SurveyMonkey.

### 3.4 Procedure and Study Design

At the beginning of the study session, participants completed an informed consent form and were given an overview of the study. After filling out a demographic questionnaire, participants completed a practice round consisting of 4 trials and 4 blocks with an icon set not used in the main study. They then completed 5 blocks of 9 trials for each of the five interfaces. The study followed a within-participant design, with the interfaces counterbalanced using a Latin square model. After each interface, participants completed NASA-TLX [35] questionnaires; after all interfaces, participants answered final questions about their preferences. Last, they reported their strategies for remembering target locations.

The study used a within-participants design with three factors (*meaning*, *shape distinctiveness*, and *colour distinctiveness*) that were used for a series of planned comparisons. The dependent measures were completion time, hover amounts, errors, and subjective responses. Our main hypotheses were:

- H1: Increased *colour distinctiveness* will reduce completion time and hover amounts *(Abstract and Concrete vs. Abstract+Colour and Concrete+Colour)*;
- H2: Increased *meaning* will reduce completion time and hover amounts *(Abstract and Abstract+Colour vs. Concrete and*

*Concrete+Colour)*;

- H3: Increased *shape distinctiveness* will reduce completion time and hover amounts *(Mixed vs. Concrete+Colour)*.
- H4: Increasing both *colour distinctiveness* and *shape distinctiveness* will lead to a larger reduction in completion time and hover amounts *(Mixed vs. Concrete)*.

## 4 STUDY 1 RESULTS

For all studies, we report the effect size for significant RM-ANOVA results as general eta-squared: $\eta^2$ (considering .01 small, .06 medium, and >.14 large [18]), and Holm correction was performed for post-hoc pairwise t-tests.

### 4.1 Completion Time

Completion time was measured from the appearance of a word cue to the selection of a correct icon; no data was removed due to outlying values. Mean completion times for the five icon sets are shown in Figure 5.

Our first planned comparisons (H1 and H2) involved the effects of colour distinctiveness and meaning. A 2x2x5 RM-ANOVA (*Meaning X Colour Distinctiveness X Block*) showed effects of *Meaning* ($F_{1,19}$= 89.60, $p$ <0.0001, $\eta^2$= 0.54) and *Block* ($F_{1,19}$= 336.88, $p$ <0.0001, $\eta^2$= 0.71) on completion time, but no effect of *Colour Distinctiveness* ($F_{1,19}$= 2.99, $p$= 0.10). There were no interactions between the factors (all $p$ >0.10).

Follow-up tests for *Meaning* showed significant differences (all $p$ <0.05) between the concrete icon sets (Concrete and Concrete+Colour) and the abstract sets (Abstract and Abstract+Colour). Follow-up tests for *Block* showed differences between each successive pair except blocks 3 and 4.

Our third planned comparison (H3) used the Mixed and Concrete+Colour conditions to see whether shape distinctiveness would improve performance in icon sets that are already distinctive in terms of colour. However, a one-way ANOVA showed no difference ($F_{1,19}$= 0.086, $p$= 0.77). Our fourth comparison (H4) used the Mixed and Concrete interfaces to see whether having two distinctive visual variables would improve performance (i.e., Mixed is more differentiable both in terms of colour and shape than Concrete). However, once again a one-way ANOVA showed no difference ($F_{1,19}$= 0.03, $p$= 0.86).

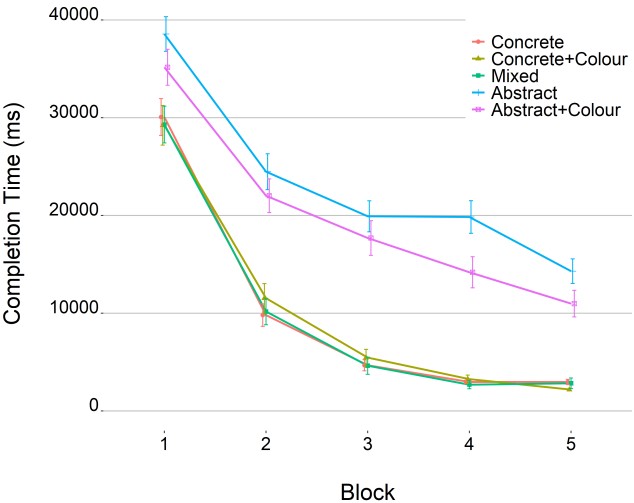

Figure 5: Mean trial completion time, by interface (±s.e.).

### 4.2 Hovers

We measured the number of hovers (where the participant held the mouse for 300ms over a target, showing the name) as a more sensitive measure of progress through the stages of cognitive, associative, and autonomous performance. As a participant moves from the cognitive to the associative stage, there should be a reduction in the number of icons that they need to inspect. Mean hovers per trial are shown in Figure 6. Results are very similar to those reported above for completion time: a 2x2x5 RM-ANOVA (*Meaning X Colour Distinctiveness X Block*) showed effects of *Meaning* ($F_{1,19}$= 117.5, $p$ <0.0001, $\eta^2$= 0.66) and *Block* ($F_{1,19}$= 353.65, $p$ <0.0001, $\eta^2$= 0.65) on number of hovers, but no effect of *Colour Distinctiveness* ($F_{1,19}$= 4.36, $p$= 0.051) (H1 and H2). There were also interactions between *Meaning* and *Colour* ($F_{1,19}$= 5.61, $p$ <0.05); as shown in Figure 6, the Abstract+Colour condition has fewer hovers than Abstract, whereas Concrete+Colour has more hovers than Concrete.

Follow-up tests for *Meaning* again showed significant differences (all $p$ <0.05) between both concrete icon sets (Concrete and Concrete+Colour) and both abstract sets (Abstract and Abstract+Colour). Follow-up tests for *Block* showed differences between successive pairs except for blocks 3 and 4.

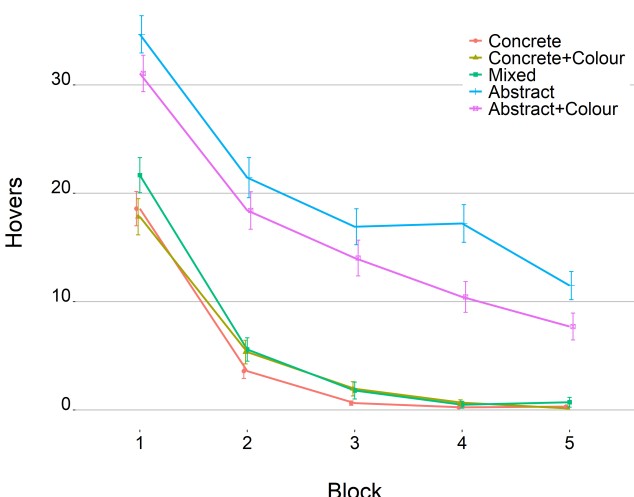

Figure 6: Mean hover amounts, by interface (±s.e.).

### 4.3 Errors

We measured errors as the number of incorrect clicks before choosing the correct item. In some trials, participants clicked instead of hovering, leading to unusually high numbers of errors; we therefore removed 32 outliers out of 4500 total trials that were more than 3 s.d. from the mean. Overall errors were low (an average of 0.032 errors per click). A 2x2x5 RM-ANOVA (*Meaning X Colour Distinctiveness X Block*) to look for effects on errors showed a main effect of *Block* ($F_{1,19}$= 12.2, $p$ <0.05, $\eta^2$= 0.046) and a main effect of *Meaning* ($F_{1,19}$= 5.16, $p$ <0.05, $\eta^2$= 0.18). Follow-up t-tests showed that abstract icons had a significantly ($p$ <0.05) higher error rate (0.048 errors per trial) than concrete icons (0.018 errors per trial).

### 4.4 Subjective Responses and Comments

We used the Aligned Rank Transform [82] to perform RM-ANOVA on the NASA-TLX responses. As shown in Figure 7, mean scores of all TLX measures followed a trend similar to completion time. We found significant effects for all subjective measures. Follow-up t-tests revealed significant differences (all $p$ <0.05) between the two

| Study | Condition | Meaning | | | Shape Distinctiveness | | | Colour Distinctiveness | | |
|---|---|---|---|---|---|---|---|---|---|---|
| | | Meaningless | Contextual | Familiar | None | Medium | High | None (Monochrome) | Medium (Colour) | High (Multi-colour) |
| S1 | Concrete | | x | | | x | | x | | |
| S1 | Concrete+Colour | | x | | | x | | | x | |
| S1 | Mixed | | x | | | | x | | | x |
| S1 | Abstract | x | | | | x | | x | | |
| S1 | Abstract+Colour | x | | | | x | | | x | |
| S2 | Square | x | | | x | | | x | | |
| S2 | Square+Colour | x | | | x | | | | x | |
| S2 | UnfamiliarShape | x | | | | | x | x | | |
| S2 | FamiliarShape | | x | | | x | | x | | |

Table 1: Icon properties of the interfaces in Study 1 & 2.

conditions with abstract icons (Abstract and Abstract+Colour) and the three conditions with concrete icons (Concrete, Concrete+Colour and Mixed) for every measure except physical effort. Significant effects were also found (all $p < 0.05$) in physical effort between Abstract and the three conditions with concrete icons as well as between Abstract and Abstract+Colour in perceived success.

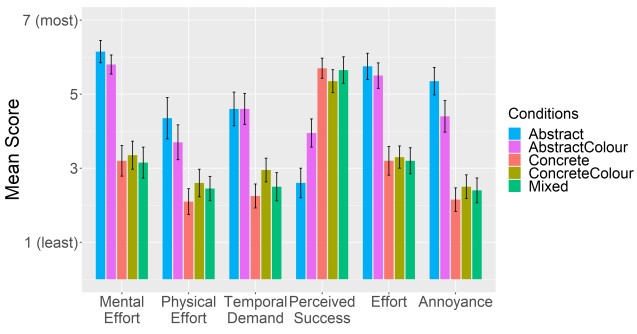

Figure 7: Mean NASA-TLX questions responses for Study 1 (±s.e.).

Overall, participants preferred both Mixed and Concrete+Colour conditions. They also perceived them as the easiest and fastest conditions where they made the least errors. Results of the preference survey are summarized in Table 2.

| | Easiest | Fastest | Fewest Errors | Preference |
|---|---|---|---|---|
| Abstract | 0 | 0 | 0 | 0 |
| Abstract+Colour | 3 | 3 | 3 | 3 |
| Concrete | 3 | 3 | 3 | 4 |
| Concrete+Colour | 6 | 7 | 7 | 7 |
| Mixed | 8 | 7 | 7 | 6 |

Table 2: Summary of preference survey results.

Participants used a variety of techniques to learn and retrieve the icons. Eight participants stated that they relied on icon meaning and attempted to find a story or link to use as the basis for their memory: for example, one participant said *"I tried to make a connection between the icon and the word."* Ten participants focused on remembering the spatial locations (at different levels of specificity); one stated *"[I recalled] the location of an icon if it was in the first, middle, or end [of the toolbar]."* Nine participants also commented on the value of shape distinctiveness. For example, a participant said *"If I had a good grasp of the icon's shape, it was easier to mentally place it in on the screen and find it again."* The same participant

reported a challenge with the less-distinctive icon sets: *"I couldn't properly grasp a unique shape [in Abstract or Abstract+Colour], it became very difficult to mentally recall its position."* Finally, six participants also used the colour of icons; one stated *"colour added an additional element for memory."*

## 5 STUDY 2 METHODS

Study 1 suggested that colour did not improve learnability, and that icons with concrete imagery were substantially easier to learn. In Study 2, we expand on these results and go into more detail on two questions: first, whether colour improves learning when it is the *only* visual variable (i.e., the icons have no shape differentiability at all); and second, whether it is the differentiability of an icon's shape or the meaningfulness of the image that assists learning.

Study 2 followed a similar method to study 1, but with two alterations. To reduce the overall time needed for the session, we reduced the number of targets from nine to seven, and the number of blocks from five to four (Study 1 showed clear learning effects within four trial blocks, see Figure 5). All other elements of the study method, procedure, and apparatus were identical to Study 1.

### 5.1 Pre-Study to Choose Number of Colours

Study 1 used 12 colours, a larger number than is recommended for mapping tasks by visual design guidelines. To determine a suitable number of colours, we carried out a small pre-study comparing learning rates with four [19], eight [49], and twelve colours. Similar to study 1, three interfaces were designed, each having 60 square icons with 5-pixel borders. In each interface, colours were distributed evenly among the icons (none repeated for neighboring icons). Participants carried out four blocks with seven targets in each interface. RM-ANOVA on completion time showed no effect of number of colours, although participant comments and literature [19, 81] generally supported four colours. Therefore, we used four colours for Study 2.

### 5.2 Interfaces

The interfaces in Study 2 used a similar spatial layout of 60 icons as in Study 1, but used four new icon sets to explore our new questions about the effects of colour, shape distinctiveness, and familiarity.

*Square.* The *Square* interface's icons were identical squares with a grey 5-pixel border. These icons have no colour differentiability, no shape differentiability, and no meaning. Therefore, the only way that participants could remember the correct icon was by memorizing its spatial location.

*Square+Colour.* The *Square+Colour* interface used the same square shapes as *Square* for all icons, but the icons were coloured with one of red, green, brown, or blue. Colours were evenly distributed across the 60 icons, and no neighboring icons repeated a colour. Colour brightness was adjusted to maximize differentiability following Arthur et al. [4]. With no shape distinctiveness in the icon

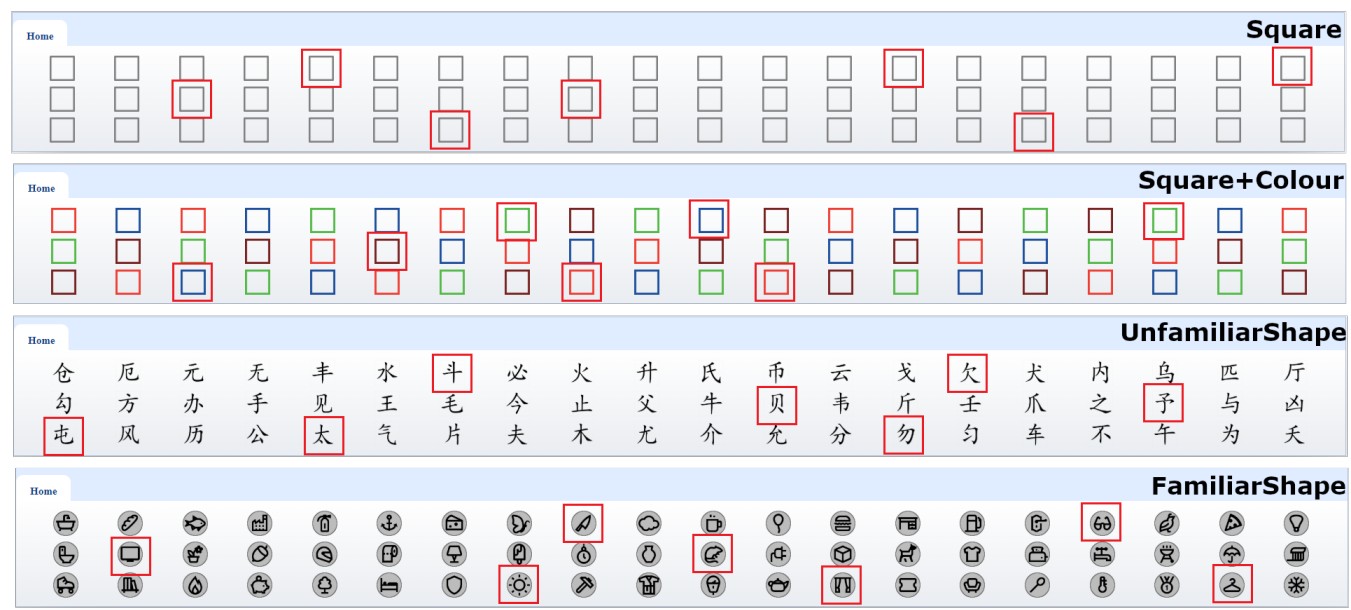

Figure 8: The four icon sets used in Study 2. Targets are outlined in red.

set, the colours provide additional landmarks for users to remember locations.

**UnfamiliarShape.** The *UnfamiliarShape* interface showed monochrome four-stroke Chinese characters as icons. These icons had high shape distinctiveness (all icons were clearly different shapes). Chinese characters are meaningful, but only if the user is familiar with them – and our participants were chosen such that none knew these characters. Therefore, this icon set had no meaning for our study.

**FamiliarShape.** The *FamiliarShape* interface used meaningful icons with imagery of recognizable real-world objects (Figure 8). Shape distinctiveness was medium, because we equalized several other visual variables such as size, line weight, and background shape (a grey circle with a 1-pixel black border).

Icons were created using GIMP. FamiliarShape's images were sourced from material.io and icons8.

### 5.3 Design

Study 2 used a within-participants factorial design with several planned comparisons. There were three factors involved in the comparisons: *shape distinctiveness* (none or high), *colour distinctiveness* (monochrome or colour), and *familiarity* (meaningless or familiar). The comparisons used different sets of conditions, as specified by our four hypotheses:

- H1: Increasing *shape distinctiveness* will reduce completion time and hover amounts *(Square and Square+Colour vs. FamiliarShape and UnfamiliarShape)*;
- H2: Increasing *colour distinctiveness* in icons with no shape distinctiveness will reduce completion time and hover amounts *(Square vs. Square+Colour)*;
- H3: Increasing *familiarity* will reduce completion time and hover amounts *(UnfamiliarShape vs. FamiliarShape)*;
- H4: Even in icons without meaning, increasing *shape distinctiveness* will reduce completion time and hover amounts *(Square vs. UnfamiliarShape)*.

### 5.4 Participants

Twenty participants who did not take part in Study 1 (sixteen women, three men, and one non-binary; ages 18-37 (mean 24, SD 5)) com-

pleted the 60-minute study, and each received a $10 honorarium. Participants had normal or corrected-to-normal vision with no reported colour-vision deficiencies, and all were highly familiar with desktop and mobile applications (up to 10 hrs/wk (3), 20 hrs/wk (3), 30 hrs/wk (6) and over 30 hrs/wk (8)). Seven participants reported primarily issuing commands by navigating GUIs with mice, eleven reported using keyboard shortcuts, one reported using both and one reported using a trackpad. Overall participants were familiar with keyboard shortcuts (1-5 shortcuts (9), 6-10 shortcuts (6), 11-15 shortcuts (3), 16-20 shortcuts (1), and over 20 shortcuts (1)). None of the participants could read Chinese characters.

## 6 STUDY 2 RESULTS

### 6.1 Completion Time

Mean trial completion times are summarized in Figure 9. No data was removed due to outlying values. We carried out analyses for each of our four planned comparisons.

First (H1), a 2x4 RM-ANOVA (*Shape Distinctiveness X Block*) showed effects of both *Shape Distinctiveness* ($F_{1,19}$= 124.22, $p$ <0.0001, $\eta^2$= 0.67) and *Block* ($F_{1,19}$= 181.67, $p$ <0.0001, $\eta^2$= 0.84) on completion time, as well as an interaction between the two factors ($F_{1,19}$= 12.44, $p$ <0.01, $\eta^2$= 0.09).

The effect of *Shape Distinctiveness*, however, must be considered in light of our third planned comparison (H3) of the familiarity of icon imagery – that is, in light of the large performance difference between the two interfaces with distinctive shapes. These interfaces (UnfamiliarShape and FamiliarShape) differ in terms of the familiarity of the icon imagery, and a one-way RM-ANOVA showed a highly significant difference between them ($F_{1,19}$= 112.24, $p$ <0.0001, $\eta^2$= 0.70). As can be seen in Figure 9, the UnfamiliarShape interface was much closer in learning rate to the two interfaces with square icons, and t-tests showed no significant differences between UnfamiliarShape and Square (p >0.1), but showed that FamiliarShape was significantly different from all three other interfaces (all p <0.001). In our results, therefore, the benefit of shape distinctiveness arose only when those shapes were both differentiable and familiar.

Follow up tests for *Block* showed significant differences between each successive pair (all $p$ <0.05). The significant interaction be-

tween *Shape Distinctiveness* and *Block* can be seen in Figure 9, where the learning curve for FamiliarShape flattens before the other conditions (because users reached expertise far earlier in this condition).

Our second planned comparison (H2) investigates the effect of colour distinctiveness in icons that have no shape differentiability (Square vs. Square+Colour). A 2x4 RM-ANOVA (*Colour Distinctiveness X Block*) showed no effect of *Colour Distinctiveness* ($F_{1,19}$= 1.62, $p$= 0.2), and no interaction with *Block* ($F_{1,19}$= 0.56, $p$= 0.46).

Our fourth planned comparison (H4) looked at whether shape differentiability alone (with meaningless icons) would improve learning. We compared the Square and UnfamiliarShape conditions using a one-way RM-ANOVA, but found no difference ($F_{1,19}$= 0.27, $p$= 0.61).

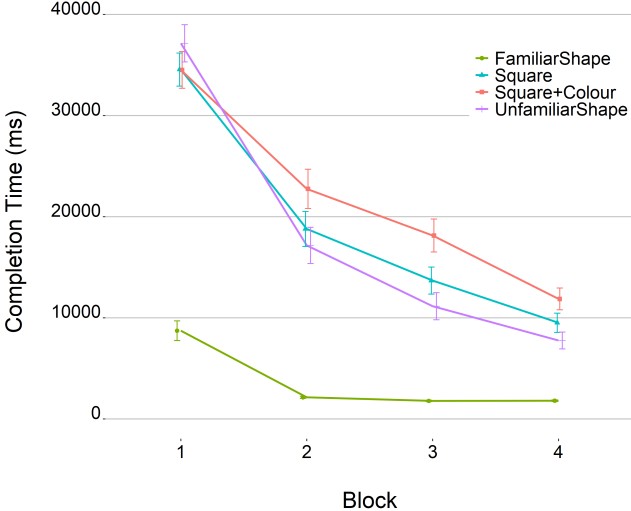

Figure 9: Mean trial completion time, by interface (±s.e.).

## 6.2  Hovers

Similar to Study 1, the results for mean hovers in Study 2 closely mirror the completion time results. RM-ANOVA (*Shape Distinctiveness x Block*) showed effects of *Shape Distinctiveness* ($F_{1,19}$= 107.02, $p$ <0.0001, $\eta^2$= 0.73), *Block* ($F_{1,19}$= 290.45, $p$ <0.0001, $\eta^2$= 0.84) as well as an interaction between the two factors ($F_{1,19}$= 24.4, $p$ <0.0001, $\eta^2$= 0.18) on hovers (H1). Follow-up tests for *Block* showed significant differences (all $p$ <0.05) between each successive pair.

As with the completion time results, the effect of Shape Distinctiveness appears to be largely due to the substantial effect of familiarity: in our third planned comparison, a one-way RM-ANOVA also showed a significant effect between UnfamiliarShape and FamiliarShape ($F_{1,19}$= 102.17, $p$ <0.0001, $\eta^2$= 0.73). T-tests also showed no significant difference between UnfamiliarShape and Square (p >0.1), but showed that FamiliarShape was significantly different from all three other interfaces (all p <0.001). Follow-up tests for *Block* showed significant differences between every successive pair except blocks 3 and 4.

In our second planned comparison (H2), a 2x4 RM-ANOVA found no effect of *Colour Distinctiveness* (p >0.15) and no interaction with *Block* ($F_{1,19}$= 0.15, $p$= 0.71, $\eta^2$= 0.002).

In our fourth planned comparison (H4), a one-way RM-ANOVA found no effect of shape differentiability ($F_{1,19}$= 0.27, $p$= 0.61, $\eta^2$= 0.006).

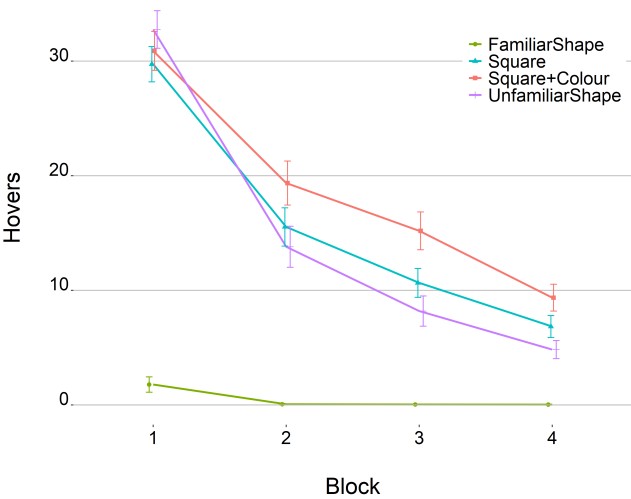

Figure 10: Mean hover amounts, by interface (±s.e.).

## 6.3  Errors

We measured errors as the number of incorrect clicks before choosing the correct item. Data from one participant (who clicked instead of hovered) was removed. For all other participants, errors were very low, with an overall average of 0.037 errors per trial. RM-ANOVA showed no main effect of any of our main factors on errors (*Shape Distinctiveness*: $F_{1,19}$= 1.42, $p$= 0.24; *Block*: $F_{1,19}$= 0.39, $p$= 0.75; *Colour*: $F_{1,19}$= 1.67, $p$= 0.21; or *Familiarity*: $F_{1,19}$= 2.47, $p$= 0.13).

## 6.4  Subjective Responses and Comments

NASA-TLX responses were analyzed after performing an Aligned Rank Transformation [82]. Data from two participants, which was incomplete, was removed. The mean effort scores shown in Figure 11 mirror the trend in the performance data, in which FamiliarShape outperformed others in all measures. RM-ANOVA showed significant effects for all subjective measures. Follow-up tests showed significant (all $p$ <0.05) differences between FamiliarShape and every other condition in mental effort, perceived success, effort, and annoyance. Overall, the FamiliarShape icons were greatly preferred - results are summarized in Table 3.

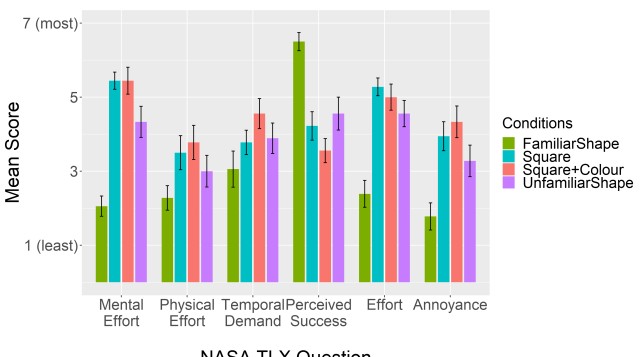

Figure 11: Mean NASA-TLX questions responses for Study 2 (±s.e.).

Participants' comments again echoed the performance results. Three participants stated that the uniformity in the Square condition was challenging; one said, *"it was really hard since everything looked the same."* Four participants also noted difficulties when attempting to use the colour information. For example, one participant

| | Easiest | Fastest | Fewest Errors | Preference |
|---|---|---|---|---|
| Square | 1 | 0 | 0 | 1 |
| Square+Colour | 0 | 1 | 0 | 0 |
| UnfamiliarShape | 0 | 0 | 1 | 0 |
| FamiliarShape | 19 | 19 | 19 | 19 |

Table 3: Summary of Study 2 preference survey results.

stated *"I tried to use colour [in Square+Colour] but it didn't work super well."* The realistic representation of targets in FamiliarShape was found to be beneficial to eight participants (e.g., *"remembering the picture of each object, and my brain just brought me to where it was"*). Finally, six participants stated that the distinct shapes of the UnfamiliarShape condition provided a connection that helped them to remember targets: for example, one participant reported that one target's icon *"looked like a bent cross"*, making it easier to remember the location.

## 7 DISCUSSION

Our two studies provided the following findings:

- Colour distinctiveness did not improve learning in either study.
- Adding multiple distinctive variables (colour and shape) also did not improve learning.
- Shape distinctiveness when coupled with meaning substantially improved learning, but shape distinctiveness on its own was not effective.
- Participant strategies suggested that they primarily try to search by meaning rather than visual characteristics.

In the following paragraphs, we consider explanations for these main results, limitations to our findings, and directions for future research.

### 7.1 Explanation for Results

#### 7.1.1 Colour distinctiveness does not improve icon learning

Colour distinctiveness did not reduce completion time or hover amounts in either study, even when it was the only visual variable available (Study 2). One main reason for this finding is that many participants apparently did not use the colour cues, and instead only searched by meaning and spatial location – participants often reported creating and connecting stories to icons to remember them rather than using colour as a visual landmark. However, participant comments suggest that at least a few people attempted to use colour information – some participants would search by colour first in target selection, allowing them to narrow down the target set (e.g., searching for red icons with two crossing lines reduces the number of icons that must be searched). But in many cases, attempts to use colour appeared to be unsuccessful. One reason for colour's ineffectiveness may be that the colour cues interfered with one another, reducing the value of colour as a landmark. That is, because all icons were coloured, remembering only that an icon was "beside the blue one" did not uniquely identify a target (because there were several blue icons) [38, 41]. It is, however, possible that if there were fewer coloured icons, colour might be a more effective landmark – in studies of artificial landmarks, for example, having only grey-coloured obvious landmarks significantly improved performance [78] in a similar selection task. It is also possible that colour interfered with participants' ability to see differences in the abstract shapes used in Study 1; that is, the colours used in the Abstract+Colour condition may have reduced contrast and thus reduced any potential effect of shape distinctiveness [44, 55].

#### 7.1.2 Shape distinctiveness was only effective with meaning

When icons had even a contextual level of meaning, we observed that participants would visually search using meaning as a memory cue; and when meaning was available, participants tended to disregard the landmarks created by differences in the icons' visual presentation. In icons with meaningless imagery, participants needed to rely more on absolute spatial memory – and without pre-existing knowledge of the icon mappings, participants had to find a prompted icon by laborious visual search (hovering one by one). In addition to improving performance in the early stages of learning, it was also clear that meaningful icons had a similar learning curve to the other conditions, implying that these conditions also allowed users to switch to location-based retrieval. Our findings confirm previous guidance about designing icons with clear meaning to help user navigation of an interface (e.g., [7, 25, 39, 40, 44, 57]), although our results extend this guidance to the value of meaning for longer-term learning of an interface as well. In contrast, Study 2 showed that shape distinctiveness without meaning did not improve learning, and the reasons for this condition's poor performance are similar to that of the colour conditions: namely, interference between similar-looking shapes may have prevented a shape's differentiability from being useful as a landmark. As with colour, shape may still be useful as a landmark if there are fewer shapes that have more noticeable differences.

### 7.2 Design Implications and Generalizing the Results

Our results suggest that user learning of an interface is not hindered by the lack of visual distinctiveness in 'flat' and subtle icon designs, and also clearly show the value of using concrete and familiar imagery. Therefore, designers can use flat and subtle icon styles without compromising memorability, as long as meaning is clearly conveyed. We note, however, that there are other potential factors in the use of flat icons that should be considered in addition to meaning (e.g., whether users can tell that an on-screen object is in fact a clickable icon). Our results also raise the question of what designers should do in situations where they must create icons for commands or concepts that do not have obvious visual representations. The frequency with which we saw the "memory hook" strategy in our studies (i.e., looking for a connection between the image of the icon and the associated command) suggests that concrete imagery – even if not a direct representation of the underlying concept – may enable learning better than simply using distinctive visual variables. As suggested above, however, it may be that the value of colour or shape distinctiveness as landmarks could be improved, a topic we will consider in future studies.

### 7.3 Limitations and Future Work

There are several ways in which our studies could not exactly replicate various factors of real-world interface learning, and these suggest possibilities for future research. First, we plan to test the idea mentioned above that colour and shape differentiability could be more effective if there are fewer items in the set that are different, thus providing a better anchor for spatial learning. One implementation would involve strategically placed icons that are designed to catch the user's attention (using colour or shape) within the toolbar; these icons could anchor memory and potentially improve learning.

Second, a limitation in our studies was the short time available for learning – users typically learn an interface in a much slower fashion, and in the context of real tasks. In addition, we tested only immediate recall, not retention after a time period, and we did not test transfer from the training task back to a real-world task with the interface. We plan retention and transfer phases in our future studies.

## 8 CONCLUSION

Icons are a ubiquitous mechanism for representing commands in an interface [7], and learning the icons in an interface is a major part of becoming an expert with that system. Despite the prevalence of icons, toolbars, and ribbons, however, little is known about the effects of icon design on learnability. We carried out two studies to test whether differentiability in two visual variables – colour and shape – would improve learning of icons in a 60-item toolbar. Our results showed that our manipulations of these variables did not have significant effects on learning or performance, and that the concreteness and meaning of the icon's imagery was far more effective in helping users learn and recall targets. Our studies provide new empirical evidence for existing guidelines that suggest an icon that are contextual or familiar will be more learnable and easier to navigate. This work increases understanding of how users learn new icons and the relative roles that visual variables and cognitive factors play in users' spatial learning and expertise development.

## ACKNOWLEDGMENTS

We would like to thank our participants and the anonymous reviewers for their feedback. This work was supported by the Natural Sciences and Engineering Research Council of Canada.

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
