# OpenReview forum: "Effects of Visual Distinctiveness on Learning and Retrieval in Icon Toolbars"
_graphicsinterface.org/Graphics_Interface/2020/Conference — GI 2020_

### Official Review · AnonReviewer1 · 2020-04-20
**Recommending accept**

**Rating:** 8
**Confidence:** 4

**Review:**

This paper investigates how the visual distinctiveness of icons in toolbars influences the speed with which users can learn and subsequently retrieve the locations of icons. The results suggest that color and shape distinctiveness in themselves are not very helpful with learning and retrieval of icons, particularly compared to icons expressing a meaning congruent with their associated commands.

Overall, I thought this was an interesting paper to read, and I am recommending that it be accepted. I think the work is well motivated by the debate about minimalist visual designs for icons; the studies are well designed to test the effects of different kinds of visual distinctiveness; and the paper is well written, with good justifications for the methods used, and a clear analysis and presentation of the study results. I think the biggest weakness of this work is that it shows that visual distinctiveness doesn't really matter that much, but there's a value in publishing negative results.

Suggestions for revisions:

- The paper enumerates the main hypotheses (H1, H2, …) in the method section for each study, and I was expecting the results section to revisit these hypotheses with a clear statement of how they were confirmed or rejected. Doing so would more clearly communicate to the reader the results of the study.

- In the charts showing trial completion time and hover amounts, it would be good to add a slight dodge in the x-axis, so the points with error bars do not overlap one another and can be more easily compared.

- On page 6, there is a typo "Our fourth comparison (H4) compared Mixed and Concrete to see whether having two distinctive visual variable would improve performance (i.e., Mixed is more differentiable both in terms of colour and shape than Concrete)." should be "variables".

---

### Official Review · AnonReviewer2 · 2020-04-21
**Interesting studies and findings, but some concern with regards to task design**

**Rating:** 7
**Confidence:** 3

**Review:**

# Summary
This paper presents two studies that look at how visual distinctiveness (color and shape) and meaning (meaningless, contextual, and familiar) impact the learnability of an interface. In the studies, participants are asked to recall a set of target icons for each of a number of icon sets with different meaning, color distinctiveness, and shape distinctiveness.

# Overview
At a high level, the paper studies an interesting and relevant topic (to more than just icon design) and has interesting findings (including some rather surprising). It is well-written, has helpful figures/tables, and is thorough in its structure and study design. I appreciate the details such as considering how to determine a suitable number of colors for an icon set. My main concern is that the task seems to be slightly far from the real-world task (which the authors also note in their limitations), but makes me wonder a bit if the results would be able to be replicated in a more realistic scenario. I would encourage the authors to discuss why this task is a reasonably proxy for the task of using icons in software.

# Task
Continuing the discussion of the task – I think what concerns me the most is the density of icons being presented to the participant all together in one location on the screen – this isn’t a common design in current interfaces. As the authors note, it also results in not really having any “landmarks” because the visual differences weren’t enough for them to be an anchor for the participants. I also think the description of the tasks/procedure were maybe the more confusing sections in the otherwise very clear paper, and might be worth going back to clarify a bit. I wondered if the positions of the icons (and targets) were randomized between participants? My impression is no, which is slightly concerning as it is possible that certain target locations might’ve been easier to remember.

# Meaning and Distinctiveness
The authors provided a very interesting breakdown of types/levels of meaning and shape/color distinctiveness. Were the visual distinctiveness levels defined by the authors or from related work? Is there a conventional way to determine these levels (eg. how much difference in shape is counts as high versus medium distinctiveness)? While the selected icon sets do seem to fall clearly into the categories as defined in Table 1, such a dictionary of levels would be more useable in different future scenarios if this were further clarified.

# Results
Results were clearly presented – the hypotheses helped make it easier to process the statistics. I appreciated that authors used hover as a proxy for proficiency of the system rather than just speed (in particular given my concern with the task design). I would’ve been interested in more discussion of qualitative feedback to gain more insight into why some of these results occurred. For instance, is there a point at which too much visual difference (Mixed) is distracting and makes the icons harder to recall? Or when participants selected which sets they preferred, what qualities influenced that choice. In the qualitative comments, please also add numbers for more context, eg. “Several (#) participants stated that they…”

# Related Work
The paper does a thorough job of covering all the important related topics for to motivate their work and approach. The initial motivation discusses how the visual consistency of flat/subtle designs might make them harder for users to distinguish, and therefore harder to learn. They also state that learning commands is important because this is a way in which users can improve their performance and transition from novice to expert. Here I had two main questions as I was reading and they address both in their related work: 1) how do you define and measure visual distinctiveness? (though measurement could be more precise as I discuss above) 2) how does learning commands help improve users performance?

# Minor
- Figure/Table labels seem to have an extra space before the number
- “meaning they posses” -> “possess”
- in Subjective Responses and Comments for Study 1: “Concret, Concrete+Colour” -> “Concrete”
- Study 2 - Design, missing period after “planned comparisons”
- I believe usually the Chinese characters are referred to as “Simplified Chinese characters” and Mandarin is more in reference to the spoken rather than written language. Also, worth double-checking, but all characters might fit under this category, so you might not need to say “Kanji and Mandarin” everywhere (other than when referring to participants’ existing knowledge of the language)

---

### Official Review · AnonReviewer3 · 2020-04-22
**Effects of Visual Distinctiveness on Learning and Retrieval in Icon Toolbars**

**Rating:** 7
**Confidence:** 3

**Review:**

This paper presents the results from two studies to investigate the effect of icon distinctiveness on how people learn and retrieve icons from GUI. They found no evidence that increasing the distinctiveness of colours or shapes can improve learning, however, they also found that adding concrete imagery to icons makes them easier to learn.

The paper is well-written, the methodology is sound, and it is highly relevant for the GI community.

My only reservation for the work is the lack of clear contributions. Although this is not my area of expertise, however, it came as a surprise to me that there is little known about the effect of visual distinctiveness on GUI usability and learnability as claimed by the authors. The paper by Bateman et. Al. (2010) which shows that adding some visual embellishment improves the memorability of charts is somehow related and should be reviewed. Bateman, S., Mandryk, R. L., Gutwin, C., Genest, A., McDine, D., & Brooks, C. (2010, April). Useful junk? The effects of visual embellishment on comprehension and memorability of charts. In Proceedings of the SIGCHI conference on human factors in computing systems (pp. 2573-2582).

I would also like to see more background on the participants in this study. What is their age groups, technical experience/expertise? I do believe these factors could have some effect on the obtained results, hence it’s necessary to provide the information.

Do studies one and two involve the same set of participants? (within or between subject?)

---

### Meta-Review · Area_Chair1 · 2020-04-23

**Recommendation:** Accept
**Confidence:** 4

**Metareview:**

This paper received three high quality reviews, all of which expressed appreciation for the work and recommended that the paper be accepted. As such, my recommendation is that the paper be accepted.

Though all of the reviews were positive, the reviewers made a number of recommendations on how to improve the paper. All of these are minor and would be easy to do in the revision cycle, so I recommend the authors integrate them into the submission. The authors can check the individual reviews for details, but a short summary of the recommended changes are as follows:

- R1 suggests some text to revisit the enumerated hypotheses (H1, H2, …) in the results or discussion sections.

- R1 made some small recommendations on presentation.

- R2 asked for more detail or justification on why the task is a reasonable proxy for the task of using icons in software.

- R2 asked whether the positions of the icons (and targets) were randomized between participants.

- R2 asked if there was any prior research that informed the breakdown of types/levels of meaning and shape/color distinctiveness. If there is, it would be good to mention and cite them.

- R2 asked if there was more qualitative feedback from participants, which might give deeper insights into their behavior, and also asked that the number of participants citing particular themes in qualitative components should be added (e.g., "Several (#) participants stated that…"

- R3 expressed surprise that little is known about the effect of visual distinctiveness on GUI usability and learnability, and recommended a few papers that might be worth reviewing and potentially integrating into the Related Work, or used to further contextualize the paper's findings.

- R3 asked for more details on the background of the participants in the study, and whether the same participants were used in both studies.

---

### Decision · Program_Chairs · 2020-04-25

Accept